# Eight New *Sedum* Plastomes: Comprehensive Analyses and Phylogenetic Implications

**DOI:** 10.3390/genes16070761

**Published:** 2025-06-28

**Authors:** Liying Xu, Shiyun Han, Yingying Xiao, Mengsa Zhang, Xianzhao Kan

**Affiliations:** 1School of Horticulture and Landscape Architecture, Wuhu Institute of Technology, Wuhu 241006, China; xuly@whit.edu.cn; 2Anhui Provincial Key Laboratory of the Conservation and Exploitation of Biological Resources, College of Life Sciences, Anhui Normal University, Wuhu 241002, China; hansy@ahnu.edu.cn (S.H.); xiaoyingying@ahnu.edu.cn (Y.X.); zmshhh@ahnu.edu.cn (M.Z.); 3The Institute of Bioinformatics, College of Life Sciences, Anhui Normal University, Wuhu 241002, China

**Keywords:** *Sedum*, *Crassulaceae*, *Saxifragales*, plastome, inverted repeats contraction and expansion, phylogeny

## Abstract

**Background:** *Sedum*, with the largest number of species in the family *Crassulaceae*, is a taxonomically complex genus and an important group of horticultural plants within this family. Despite extensive historical research using diverse datasets, the branching patterns within this genus and the family remain debatable. **Methods:** In this study, we conducted sequencing and comparative analyses of plastomes from eight *Sedum* species, focusing on the diversities in nucleotide, microsatellite repeats, putative RNA editing, and gene content at IR junctions. The phylogenetic inferences were further conducted at the order level—*Saxifragales*. **Results:** Our IR junction analyses of the eight investigated *Sedum* species detected a unique 110 bp IR extension into *rps19*, a feature highly conserved across *Crassulaceae* species, indicating a remarkably family-specific pattern. Additionally, we obtained 79 PCGs from 148 *Saxifragales* species and constructed a phylogenetic tree using a larger set of plastomes than in previous studies. Our results confirm the polyphyly of *Sedum* and reveal that *S. emarginatum* is more closely related to *S. makinoi* than to *S. alfredii*, which is sister to *S. plumbizincicola*. Furthermore, we also performed analyses of codon usage, putative RNA editing sites, and microsatellite repeats. **Conclusions:** These findings and the generated sequence data will enrich plastid resources and improve understanding of the evolution of *Sedum*, *Crassulaceae*, and *Saxifragales*.

## 1. Introduction

*Sedum*, the most species-rich genus of the family *Crassulaceae* (*Saxifragales*), consists of approximately 470 divergent species [1], accounting for one-third of the whole family [2]. This genus has a long and complicated taxonomic history, with *Sedum* morphology being particularly difficult to distinguish [2]. At first, *Sedum* was placed in the subfamily Sedoideae according to a previously widely accepted classification [3,4]. Later, with improvements in phylogenetic inference methods, it was reclassified into Sempervivoideae [5,6], which is now widely accepted. This genus has been widely perceived as complex in taxonomy. *Sedum* was first considered polyphyletic by Van Ham in 1998 [7]. Over time, many research efforts have been devoted to exploring the phylogenetic affinities within *Sedum* [8,9,10,11,12]. However, the polyphyletic nature of this genus consistently poses an obstacle to elucidating its internal branching patterns. To date, based on limited published data, the evolutionary relationships with this genus remain controversial and poorly understood [1,2,13], requiring more data and further investigation.

The plastid plays a pivotal role in photosynthesis as well as plant growth and the synthesis of essential metabolites [14]. Although linear plastomes have also been reported [15,16], land plant plastomes generally possess remarkably conserved structures, and mostly display a circular quadripartite structure [17,18,19,20]. The plastome not only evolves slowly [14,19] but also is mostly single-parent inherited, making it an ideal candidate for phylogenetic research [14,21,22,23]. More significantly, our previous works have revealed several plastomic features possessing significant phylogenetic implications, such as gene content adjacent to IR junctions [24,25] and unique codon usage patterns [26,27]. Currently, despite its sufficiently meaningful status in *Crassulaceae* as well as high research value and taxonomic complexity, the available plastome resources of *Sedum* remain surprisingly limited. Therefore, a comprehensive plastomic investigation and phyloplastomic inferences based on more samples are urgently required. These will shed light not only on the internal relationships of *Sedum* and *Crassulaceae* but also their plastomic evolution.

Here, we newly sequenced the complete plastomes of eight species of *Sedum* and report them herein. This work can supplement existing data resources and will be advantageous in terms of gaining insight into the evolution of this genus with relatively wide sampling. Notably, this study aimed to (1) explore the plastomic evolution and inter-species diversity of *Sedum*, (2) compare the characteristic patterns of gene content at the IR boundaries in *Sedum* and the family *Crassulaceae*, and (3) promote the understanding of phylogenetic affinities under a rather wide sampling scale.

## 2. Materials and Methods

### 2.1. Sample Material, DNA Extraction, and Sequencing

Fresh leaf samples from eight *Sedum* species, namely, *S. alfredii*, *S. burrito*, *S. clavatum*, *S. dasyphyllum*, *S. furfuraceum*, *S. hernandezii*, *S. makinoi*, and *S. palmeri*, were collected from the greenhouse of Anhui Normal University (Wuhu, Anhui, China). All eight species are included in the largest major clade of *Crassulaceae*, the Acre clade. We used the CTAB method for genomic DNA extraction. A TruSeq DNA PCR-Free Library Prep Kit (Illumina, San Diego, CA, USA) was used to construct the library, and then sequencing was performed through an Illumina HiSeq X Ten (Novogene Co., Ltd., Beijing, China) with a 150 paired-end strategy.

### 2.2. Genome Assembly and Annotation

After the quality checking of raw reads with FastQC v.0.12.1 [28], the sequenced high-quality clean reads were assembled in GetOrganelle v.1.7.7.1 [29], taking the plastome of *S. plumbizincicola* (MN185459.1) [30] as a reference. The generated assembly results were then annotated using GeSeq v.2.03 [31], and the junctions of quadripartite regions were verified. The annotation results were further checked and modified manually. We subsequently visualized these plastomes using Chloroplot [32].

### 2.3. Codon Usage, Putative RNA Editing Sites, and IR Junction Analysis

In the codon usage analysis, we used MEGA 12 [33] to determine the relative synonymous codon usage (RSCU) values of protein coding genes (PCGs). For the presumptive RNA editing sites, PREP-Cp [34] was applied with a 0.8 cutoff. An overall comparison of the quadripartite boundary sites among the eight plastomes was performed with IRscope [35], which was further corrected and plotted manually. Moreover, we conducted larger-scale IR junction analyses, including six *Crassulaceae* plastomes from different genera and eight *Saxifragales* plastomes covering multiple families (marked in Appendix A).

### 2.4. Microsatellite Repeat Analysis

With the removal of one copy of the IR regions, we used MISA-web (a tool for microsatellite prediction) [36] to investigate potential simple sequence repeats (SSRs) in the eight plastomes. The criteria for SSR identification included thresholds of 8, 4, and 3 repeats for mono-, di-/tri-, and tetra-/penta-/hexanucleotide SSRs, respectively.

### 2.5. Phylogenetic Analysis

To gain insights into the phylogeny of *Sedum* and *Crassulaceae*, we conducted phylogenetic inferences using a broad sampling scale across the order *Saxifragales*. We downloaded 140 plastomes of *Saxifragales* species from the NCBI database, along with 2 Vitales species as outgroups (see Appendix A for details). Generally, we extracted 79 PCGs from a total of 150 species and multiply aligned them with MAFFT v.5.3 [37]. To infer phylogeny of *Saxifragales*, we employed two different methods for tree reconstruction: (1) RAxML v.8.2.12 [38] using the maximum likelihood (ML) method, with the GTRCAT model and a bootstrap analysis of 50 runs and 1000 replicates; and (2) MrBayes v.3.2.6 [39] using the Bayesian inference (BI) method, calculating the best-fit models for each gene with ModelTest-NG v.0.1.7 [40], with four Markov chains running twice for 10 million generations.

## 3. Results

### 3.1. Plastid Genome Organization and Features

The assembled plastomes of eight *Sedum* species ranged in size from 148,618 to 150,714 bp, displaying typical quadripartite structures containing LSC regions (80,301–82,779 bp) and SSC regions (16,657–16,751 bp), which were flanked by a pair of IR regions (25,532–25,804 bp for single IR copy) (Figure 1). All species displayed similar GC content, with IR regions having the highest GC content, followed by SSC and LSC (Table 1). Found to be extremely conserved, the gene content exhibited strong similarities among the eight *Sedum* species. The sampled plastomes all harbored 133 genes, comprising 85 PCGs, 8 rRNAs, 36 tRNAs, and 4 pseudogenes (Appendix A). There were 17 genes containing introns, involving 12 CDSs and 5 tRNAs. Of these, 14 genes had one intron, and 2 genes (*ycf3* and *clpP*) each harbored two introns. The two 3′-end exons of the trans-splicing *rps12* gene are located within the IR region, while the 5′-end exon is present in the LSC region. Furthermore, owing to the location of the *ycf1* gene, at the IRa-SSC boundary, a truncated copy was found in the corresponding IRb region. A similar case was observed in the *rps19* gene.

### 3.2. Comparative Analyses of Nucleotide Composition, Codon Usage, and Amino Acid Frequencies

The nucleotide composition was compared among the eight *Sedum* species, showing high similarity. The eight plastomes all showed an AT bias, ranging from 62.04% to 62.25%, with a slight T-skew (AT-skew ranges from −0.00853 to −0.00708) and C-skew (GC-skew ranges from −0.01908 to −0.01745) (Table 2). In the protein coding sequences (CDSs), Thymine and Adenine tend to be remarkably uniform, with occurrences of 54.5% (position 1), 62.2% (position 2), and 70.2% (position 3) in codons. Detailed results for each plastome are shown in Appendix A. Further analysis of the amino acid frequency among the eight plastomes revealed a highly analogous pattern, in which Leucine was the most abundant amino acid, while Cysteine was detected as the rarest (Figure 2).

Within the protein coding regions of the eight plastomes, sixty-four synonymous codons were found. Compared to codons that ended with G/C, those consisting of A/T at the third position showed a higher frequency and encoded most of the amino acids. To gain better insight into the synonymous codon usage pattern, we determined the RSCU values of the eight plastomes (Figure 3, Appendix A). In this analysis, the codons ATG and TGG were found to be unbiased, and all eight plastomes preferred A/T-ending codons to G/C-ending ones. Moreover, the most frequent initial codon was investigated and found to be ATG, whereas other types were also identified in some genes, including ACG and GTG.

In addition to these conserved codon usage patterns, we discovered several interspecific heterogeneities. An innovative comparison tool, significantly variable codons (SVCs), was developed here to compare RSCU patterns among species. The SVCs represented codons with a distinct bias (preferred when RSCU > 1; unpreferred when RSCU ≤ 1) between taxa. A total of 16 SVCs were identified (Appendix A). Notably, these codons might potentially serve as unique markers for two *Sedum* species: *S*. *burrito* and *S*. *palmeri*. Nine of the sixteen SVCs were exclusively possessed by *S*. *burrito*, while *S*. *burrito* and *S*. *palmeri* shared the remaining seven.

### 3.3. Microsatellite Repeat Polymorphisms

In these eight *Sedum* plastomes, we detected abundant SSRs (129 to 150 per genome), with several variations in both repeat types and numbers. The majority of the whole SSRs across these plastomes were universally composed of mononucleotide repeats (Figure 4b). Their occurrence rates typically ranged from 68.2% to 75.2%. For instance, *S. alfredii* showed a rate of 75.2%, *S. burrito* was 71.3%, and *S. furfuraceum* was 68.2%. In contrast, penta- and hexanucleotides were rarely identified. Additionally, the overwhelming majority of these mono-SSRs in all eight plastomes were of the A/T type. As shown in Figure 4a, most SSR loci among the eight *Sedum* species were located in the LSC region, followed by the SSC and then the IR regions. Detailed information on the types and frequencies of the SSRs is displayed in Appendix A.

Despite these similarities, several variations were also found. Among all eight plastomes, *S. makinoi* contained the most trinucleotide SSRs (six, while the others only had one to two). Furthermore, no hexanucleotide SSRs were detected, except for *S. burrito*. In summary, the SSRs among the eight *Sedum* plastomes presented abundant diversity.

### 3.4. Prediction of RNA Editing Sites

RNA editing events represent an important mechanism for altering RNA sequences to ensure the essential synthesis of proteins. To the best of our knowledge, RNA editing is widespread in land plant plastids [41,42]. The analysis conducted on PREP-Cp revealed putative RNA editing sites in the protein coding sequences of the eight *Sedum* plastomes. These sites exhibited a certain level of similarity, with the most abundant RNA editing sites predicted in *ndhB* (9–11) and *rpoC2* (4–6) in all eight plastomes. Notably, one consistent pattern across all eight *Sedum* plastomes was that nearly half of the RNA editing sites (43.9–56.1%) occurred in codons encoding Serine. Of these, the most frequent amino acid conversion type was Serine to Leucine. In all these plastomes, RNA editing events mostly occurred at the second nucleotide (73.2–80.5%), followed by the first position (19.5–26.8%). Moreover, almost all the other amino acids had a single kind of conversion, with the exception of Proline, which converted to Serine via first-position editing and to Leucine via second-position editing (Appendix A).

With an overview of the specific distribution of putative editing sites in the eight plastomes (Figure 5), two remarkable results were observed. First, *S. dasyphyllum* had conspicuously more editing sites in more genes (52 sites in 21 genes) than the other species, including unique sites in *petG* and *psaI*. Second, RNA editing loci were predicted in *rpoA* in all plastomes except *S. hernandezii*.

### 3.5. IR Contraction and Expansion

A comparative analysis of IR contraction and expansion was performed on the eight *Sedum* plastomes, together with other species belonging to *Crassulaceae* (Figure 6a) and *Saxifragales* (Figure 6b). The *Crassulaceae* species showed both similarities and unique features. For instance, due to the functional *ycf1* gene occupying the IRa region, all these plastomes contain a complete *ycf1^Ψ^* (a pseudo-copy of *ycf1*) within the IRb region, extending to the SSC/IRb boundary. Furthermore, the *ndhF* gene typically extended across the SSC region (largely) and into the IRb region (partially). Overlaps of 34–59 bp were detected between the *ycf1^Ψ^* and *ndhF* genes in all *Crassulaceae* plastomes.

Strikingly, a unique feature was discovered in *Crassulaceae* plastomes when compared with representative plastomes from other families within the order *Saxifragales*. In *Crassulaceae* species, the *rps19* gene is situated at the LSC/IRb junction, with the IRb region extending 110 bp into this gene. However, our results indicate that this extension was absent in the other examined families of *Saxifragales*.

### 3.6. Phylogenetic Analysis

In addition to our eight *Sedum* species, we obtained 140 *Saxifragales* species plastomes, representing 11 of the 15 families [43], along with two Vitales taxa used for outgroup comparison. After alignment and concatenation, the sequence matrix length was 73,719 bp. Our phylogenetic analyses revealed nearly identical tree topologies between the ML and BI approaches (Figure 7). In general, the order *Saxifragales* comprised two major clades. The woody clade (containing Altingiaceae, Cercidiphyllaceae, Hamamelidaceae, and Daphniphyllaceae), alongside Paeoniaceae, clusters into a specific clade, which is the sister group to the core *Saxifragales* clade. The core clade further divides into two distinct lineages: a *Crassulaceae* + (Haloragaceae + Penthoraceae) clade sister to the Saxifragaceae alliance (comprising (*Saxifragales* + Grossulariaceae) + Iteaceae), which is in agreement with previous research based on fewer taxa [30,44] or other types of datasets [44,45].

The polyphyly of *Sedum* is well captured by our cladogram (Figure 7). The phylogenetic analysis based on the *matK* gene by Mort et al. [9] indicated that *S. clavatum* is sister to (*Pachyphytum compactum* + *S. burrito*). However, in our study, the cladogram strongly supported a closer placement of *P. compactum* with *S. clavatum* than *S. burrito* (ML bootstrap = 100 and BI probability = 1.0). Furthermore, we found, with strong support, that *S. emarginatum* was more closely related to *S. makinoi* than to *S. alfredii*. This finding differs from that of Messerschmid et al. [1], who reported *S. alfredii* as sister to *S. plumbizincicola* [30].

## 4. Discussion

In this study, we newly report the plastomes of eight species of the genus *Sedum* and present comparative plastome analyses. To gain insight into evolutionary relationships at both the genus and family (*Crassulaceae*) levels, phylogenetic analysis was performed at the order level, *Saxifragales*. Given the important role of *Sedum* in the family *Crassulaceae*, we believe that this study will contribute to expanding knowledge about its evolutionary history.

Plastomes have been widely used for phylogenetic analyses [23,46,47,48,49,50], as they usually have conserved genome structures and gene contents. However, gene loss events have also been reported in some lineages [30,51,52,53]. In this study, the eight *Sedum* plastomes all possessed 133 genes, comprising 85 PCGs, 8 rRNAs, 36 tRNAs, and 4 pseudogenes. This pattern of gene content is conserved across four other previously published *Sedum* plastomes. Furthermore, we analyzed the plastome structures and gene properties of all 148 *Saxifragales* species, covering 11 different families. The loss events of *infA* and *rpl32* were detected in all 17 plastomes of Paeoniaceae, reinforcing Ding et al.’s conclusions [30]. These specific gene losses can serve as potential phylogenetic markers in Paeoniaceae.

Codon usage bias (CUB) is a well-established ideal candidate for investigating the evolution of different organisms [54,55]. It is also widely employed to comprehend evolutionary patterns in genes [56,57]. The codon usage patterns of the eight *Sedum* species reported here exhibited a high degree of similarity, indicating strong conservation at the genetic level during plastid evolution. Regarding the RSCU values of each plastome, a consistent trend was observed in the eight *Sedum* species we investigated: they all obviously preferred codons ending with A/T over G/C. This finding supports previous reports on angiosperm plastids [23,46]. In terms of start codon selection, besides the most common one (ATG), occurrences of a few other types have been found in some plastid genes, including ACG and GTG (in *ndhD*, *psbC*, and *rps19*). Similar cases can be found in many studies [23,58,59]. Interestingly, the performance of these non-ATG start codons usually involves two strategies: using them directly or converting them back to ATG through RNA editing. Based on transcriptome data, Wang et al. [58] observed that some alternative start codons in certain genes underwent such editing events, while others remained unedited. This implies that unedited transcripts may have distinct functions. Further attention should be paid to understanding the underlying mechanisms of these editing events, with future studies investigating their potential phylogenetic implications. We additionally explored the SVCs among our eight plastomes. Remarkably, nine unique SVCs could act as potential markers for *S*. *burrito*. However, no significant phylogenetic implications were observed in SVC distributions, especially for the relationship between *S*. *burrito* and *S*. *palmeri*, which were not the most closely related sister group. This phenomenon potentially indicates that, despite several divergences observed, the codon usage of plastid PCGs is still conserved; however, using only limited samples meant that we could not clearly infer the relationships between these very closely related species. Therefore, a wider sampling scale and in-depth exploration remain crucial.

Repetitive DNA is widely recognized as a fundamental component in genomes that plays a dominant role in evolution [60]. Furthermore, because strong selection on a gene also reduces polymorphism in adjacent, genetically linked regions, the variation patterns of repetitive DNA serve as a useful tool for identifying these precise genomic loci [61]. Simple sequence repeats (SSRs) are important molecular markers that are widely used in genetic mapping, population structure, and phylogenetic study [62,63,64]. The MISA analysis identified 129 to 150 SSRs in each of the eight *Sudum* plastomes. The SSR patterns in these plastomes present both similarities and divergences. Among all of the plastomes, mononucleotide repeats harbored much larger quantities compared with other types. Regarding the repeat unit types, A/T SSRs and AT/TA SSRs comprised a substantial portion of mono- and dinucleotide repeats, respectively. The SSRs of these plastomes also exhibited species-specific variations. For example, trinucleotide repeats were much more numerous in *S. makinoi* than in the other species, while hexanucleotide SSRs were rare in *S. burrito*. Despite these variations, it is difficult to accurately differentiate these closely related species based solely on SSR length [58]. Notably, Schug et al. hypothesized that the mutation rate of SSRs may be directly proportional to the repeat unit lengths [65]. This indicates the importance of identifying variation in the mutation rates of SSRs when using them as markers for tracking evolutionary events [65].

Although the plastome structures of land plants are generally conservative, they still show variation. The most classic example of this is the contraction and expansion of the IR regions, which results in genome size variations. As a key feature in plant evolution [46,49,66], the boundaries of the IR regions are widely considered useful for illuminating evolutionary intricacies [23,67]. For this reason, comparisons should be conducted not only within genera but also at higher levels, such as families and orders. In this study, we compared our eight *Sedum* species with other species at three hierarchical levels: the genus *Sedum*, the family *Crassulaceae*, and the order *Saxifragales*. Remarkably, we discovered a unique pattern in the IR junctions of *Crassulaceae* plastomes. In all of the analyzed *Crassulaceae* species, the *rps19* gene is located at the LSC/IRb junction, and the IRb regions all extend 110 bp into this gene. This conserved pattern could serve as a potential phylogenetic marker for the *Crassulaceae* family. Moreover, Han et al. [68], in a comparison within *Saxifragales*, also exclusively discovered the same unique pattern in *Crassulaceae*
*rps19* genes. Based on this, we presume that this special characteristic might be a plesiomorphy. This evidence contributes to a better understanding of the evolutionary history and has potential use for species identification within *Crassulaceae*, after further verification using more data and samples.

Our phylogenetic analysis, based on 79 plastid PCGs, robustly demonstrated the polyphyly of *Sedum* and shed light on relationships at deeper nodes. Within *Crassulaceae*, all analyzed genera (*Rhodiola*, *Phedimus*, *Hylotelephium*, *Orostachys* and *Sinocrassula*) were strongly supported as monophyletic, with the exception of the polyphyletic *Sedum.* Messerschmid et al. [1] attributed this polyphyly to “Linnaeus’s folly”—treating *Sedum* as a hold-all genus—and pointed out that the original circumscription of *Sedum* crucially needs to be reclarified. Based on both molecular (one nuclear and three plastid gene loci) and morphological considerations, they preliminarily proposed to move all species of the tribe Sedeae into *Sedum*. However, further verification using more samples is needed to support this solution. More importantly, multiple molecular datasets are also essential, considering that phyloplastomic construction could be easily affected by plastid capture or hemiplasy. Here, compared with the previous work, the different taxonomic relationships of *S. alfredii*, *S. burrito*, *S. clavatum*, *S. emarginatum*, and *S. makinoi* that were intensively supported by our inferences may be ascribed to our larger range of species and greater scale of sequence data. It is worth noting that our findings may provide new insights into resolving the confounding phylogeny within *Sedum*.

In summary, this study is the first to report the complete plastomes of eight *Sedum* species and presents a comprehensive analysis of them. The unique pattern in the IR junctions of *Crassulaceae* may potentially provide a new tool for species identification. Furthermore, the well-resolved phylogenetic tree based on 79 PCGs from 148 taxa has clarified taxonomic relationships within the order *Saxifragales*. Taken together, this work provides a valuable genomic resource and a robust phylogenetic framework that will facilitate future evolutionary studies in this group.

## Figures and Tables

**Figure 1 genes-16-00761-f001:**
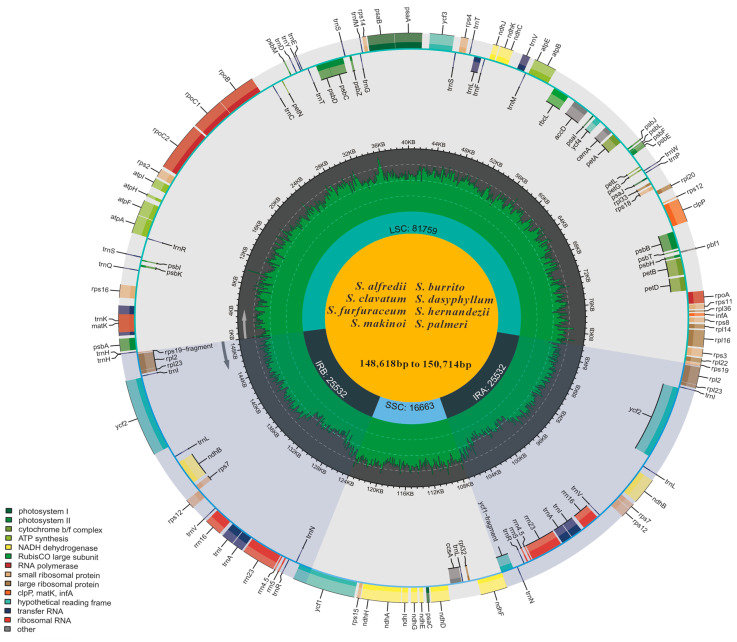
Plastome diagram illustrating eight *Sedum* species. Genes positioned inside the circle were transcribed in a clockwise direction, while those outside were transcribed in the opposite direction. Functional categories are represented by distinct colors, and pseudogenes are annotated with asterisks. The GC content is visualized through shaded segments.

**Figure 2 genes-16-00761-f002:**
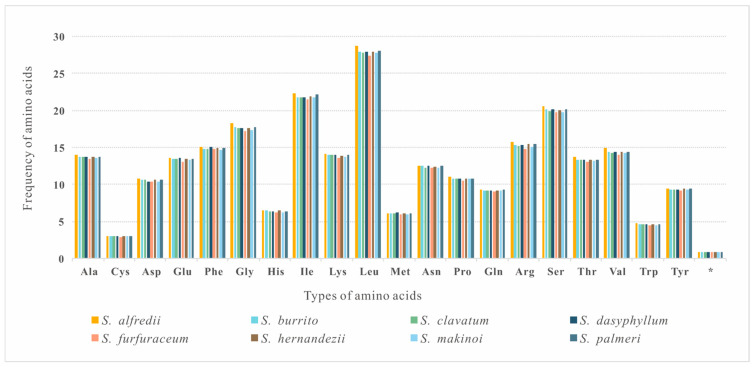
Comparison of amino acid frequency in eight *Sedum* species.

**Figure 3 genes-16-00761-f003:**
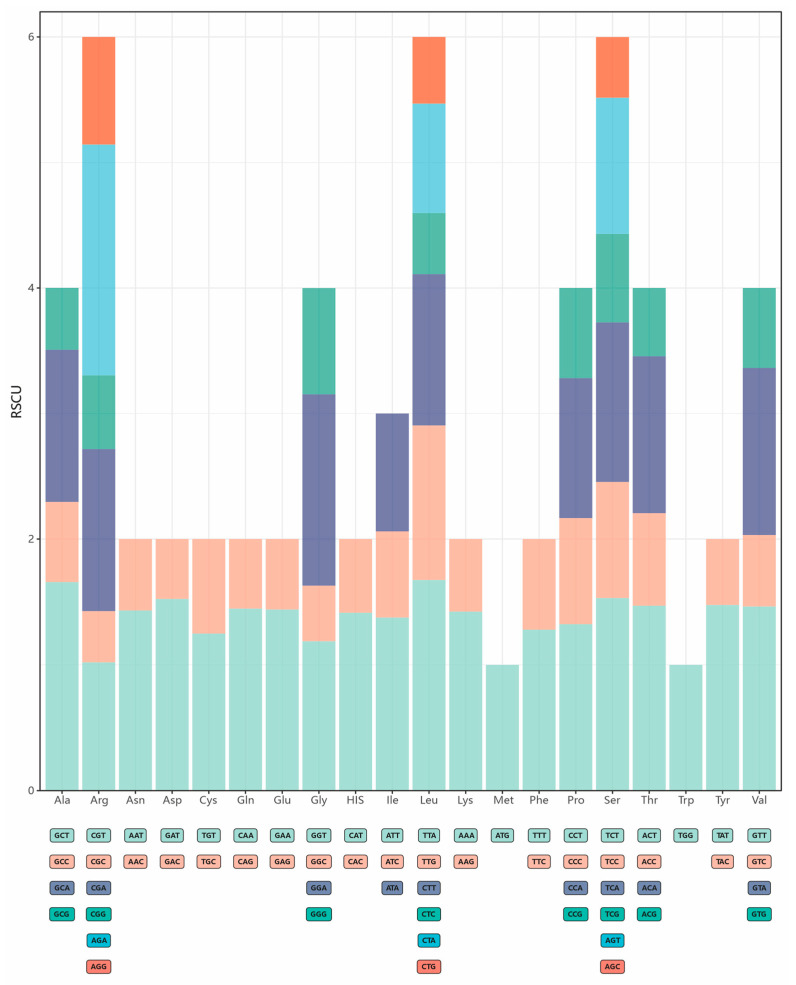
RSCU values in the plastomes of eight *Sedum* species. The specific codon index is given under the X-axis.

**Figure 4 genes-16-00761-f004:**
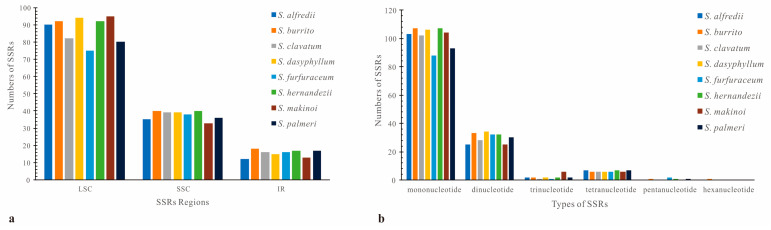
Microsatellite repeat analyses of eight *Sedum* species: (**a**) distribution of SSRs within these plastomes; (**b**) quantitative analysis of SSR types in these plastomes.

**Figure 5 genes-16-00761-f005:**
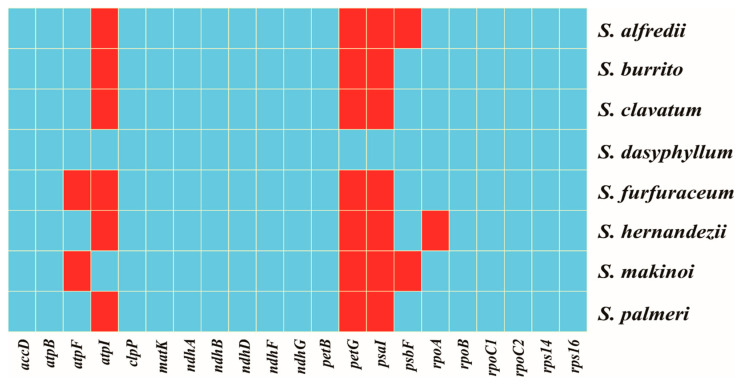
Overview of putative RNA editing sites in the eight *Sedum* plastomes. Blue squares denote the presence of RNA editing sites, while red squares denote their absence.

**Figure 6 genes-16-00761-f006:**
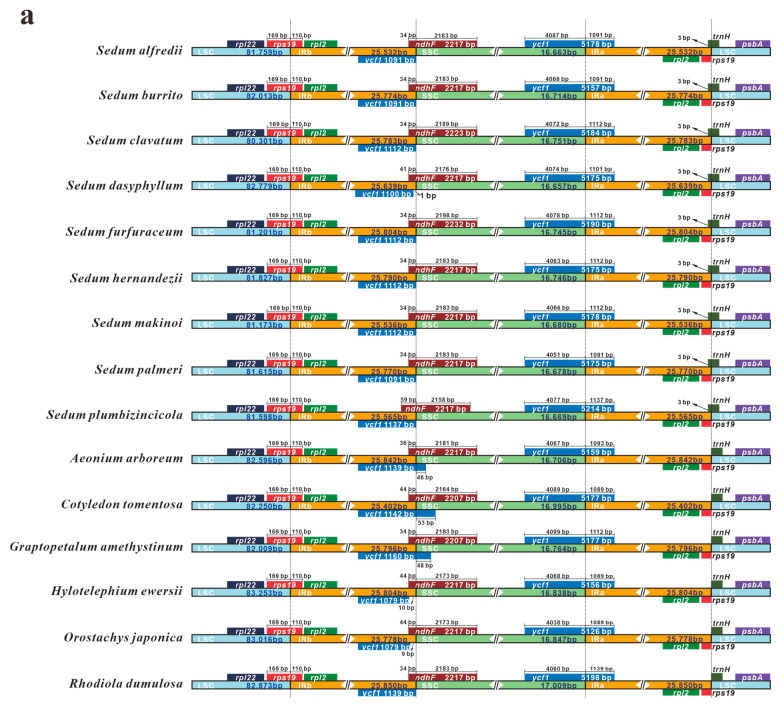
Comparative analyses of plastome boundary regions among several species: (**a**) analyses of our eight *Sedum* plastomes and seven other representatives from *Crassulaceae*; (**b**) analyses of eight plastomes of *Saxifragales* representing different families.

**Figure 7 genes-16-00761-f007:**
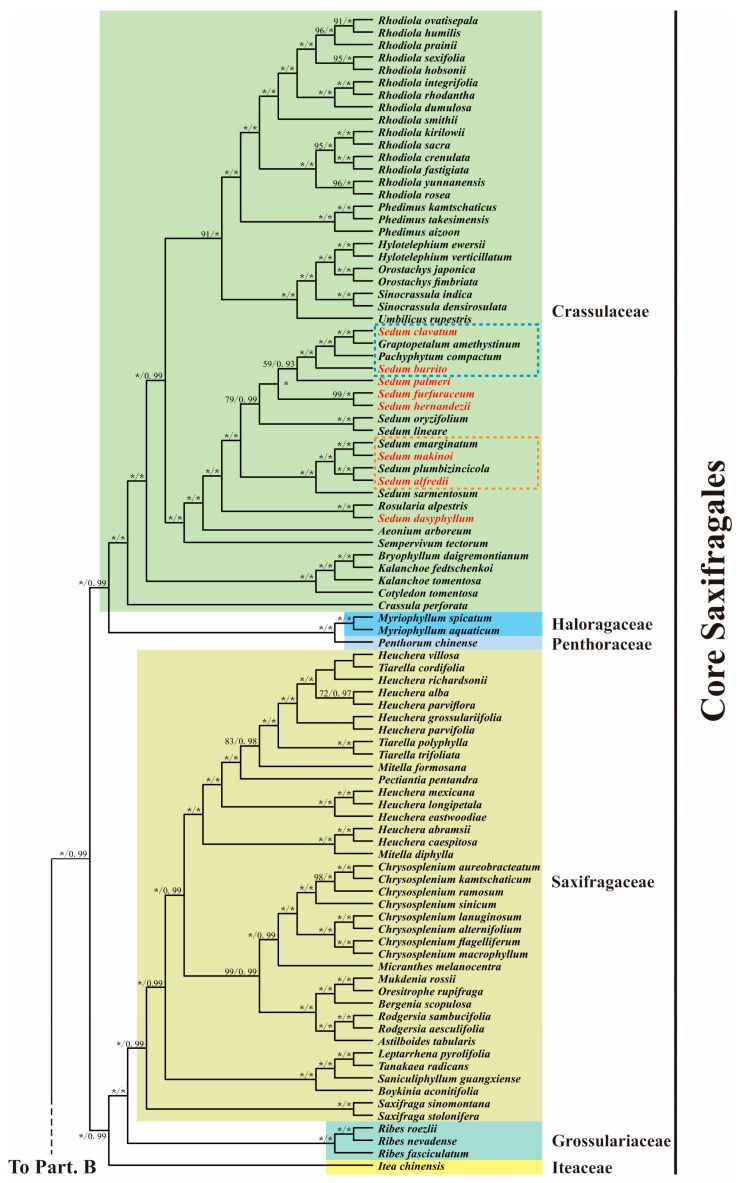
Phylogenetic cladogram reconstructed for 148 *Saxifragales* taxa using 79 protein coding genes. Branch support values, including bootstrap (BS) and posterior probability (PP), are displayed at nodes. Asterisks indicate full support [100% BS or 1.00 PP]; values below 50% BS or 0.5 PP are omitted.

**Table 1 genes-16-00761-t001:** Comparative analysis of genome features among eight *Sedum* species.

Features		*S. alfredii*	*S. burrito*	*S. clavatum*	*S. dasyphyllum*	*S. furfuraceum*	*S. hernandezii*	*S. makinoi*	*S. palmeri*
Voucher		AHNU-KL00119	AHNU-KPXY016	AHNU-KPLE014	AHNU-KPDJ018	AHNU-KPQM017	AHNU-KPLG019	AHNU-KL00214	AHNU-KPBH015
Plastome size (bp)		149,486	150,275	148,618	150,714	149,554	150,153	148,925	149,833
LSC length (bp)		81,759	82,013	80,301	82,779	81,201	81,827	81,173	81,615
SSC length (bp)		16,663	16,714	16,751	16,657	16,745	16,746	16,680	16,678
IR length (bp)		25,532	25,774	25,783	25,639	25,804	25,790	25,536	25,770
Gene number	Total	133	133	133	133	133	133	133	133
	IR regions	21	21	21	21	21	21	21	21
	CDS	85	85	85	85	85	85	85	85
	rRNAs	8	8	8	8	8	8	8	8
	tRNAs	36	36	36	36	36	36	36	36
	Pseudogenes	4	4	4	4	4	4	4	4
GC content (%)	Overall	37.7	37.9	38	37.8	37.9	37.8	37.8	37.8
	LSC	35.7	35.8	36	35.8	35.9	35.8	35.7	35.8
	SSC	31.6	32	31.9	31.9	32	31.8	31.8	31.9
	IR	43	43	43	43	43	43	43	43
	CDS	37.7	37.8	37.7	38	37.7	37.8	37.6	37.7
	rRNA	55.4	55.4	55.4	55.4	55.4	55.4	55.4	55.4
	tRNA	52.9	52.8	52.9	53	52.8	52.8	53	52.8

**Table 2 genes-16-00761-t002:** The length, base composition, and skew of the complete plastid genomes of eight *Sedum* species.

Species	Length (bp)	T%	C%	A%	G%	AT%	AT-Skew	GC-Skew
*S. alfredii*	149,486	31.37	19.23	30.88	18.51	62.25	−0.00787	−0.01908
*S. burrito*	150,275	31.29	19.30	30.83	18.58	62.12	−0.00741	−0.01901
*S. clavatum*	148,618	31.25	19.32	30.79	18.63	62.04	−0.00741	−0.01818
*S. dasyphyllum*	150,714	31.31	19.24	30.87	18.58	62.18	−0.00708	−0.01745
*S. furfuraceum*	149,554	31.29	19.32	30.79	18.60	62.08	−0.00805	−0.01899
*S. hernandezii*	150,153	31.33	19.27	30.84	18.57	62.17	−0.00788	−0.01850
*S. makinoi*	148,925	31.34	19.22	30.89	18.55	62.23	−0.00723	−0.01774
*S. palmeri*	149,833	31.34	19.28	30.81	18.57	62.15	−0.00853	−0.01876
Average	149,662	31.32	19.26	30.84	18.57	62.17	−0.00774	−0.01842

## Data Availability

All the generated data have been deposited in the GenBank database with the accession numbers PV751241–PV751248.

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
