# Peer review of "Eight New Sedum Plastomes: Comprehensive Analyses and Phylogenetic Implications"

_genes, 2025, doi:10.3390/genes16070761_

Round 1

Reviewer 1 Report

Comments and Suggestions for Authors

This manuscript contributes valuable plastome data for Sedum and sheds light on phylogenetic relationships within Crassulaceae and the broader Saxifragales. The study’s aims are clearly stated, and the findings are compelling. However, although the polyphyly of Sedum is reaffirmed, the work does not substantially advance our understanding of the evolutionary processes driving this polyphyly. In several instances, connections between data and biological interpretations could be made more explicit. My suggestions follow:

Lines 41–44: The text repeatedly emphasizes the polyphyly of Sedum, but it would benefit from a clearer distinction between confirming polyphyly and elucidating internal branching patterns. At present, these concepts are somewhat conflated, which diminishes the novelty of the findings.

Lines 56–57: While the generation of new plastomes is justified, the introduction could better clarify what specific new insights are expected beyond reaffirming polyphyly.

Lines 112–118: The identification of four pseudogenes per plastome is clearly presented, but it remains unclear whether these pseudogenes are consistent across all sampled species. Clarifying this would improve the interpretation.

Lines 143–148: Codon usage patterns are described in detail, but the section remains descriptive. It would be useful to explore whether observed codon usage variations align with phylogenetic positioning or suggest potential adaptive significance.

Lines 165–168: SSR type and count variations are noted but not analyzed. Consider discussing whether these differences might aid in distinguishing species within the genus or hold population-level implications.

Lines 179–187: While consistent RNA editing patterns are noted, the manuscript does not discuss their relevance to phylogenetic resolution or functional significance. A brief biological interpretation would strengthen this section.

Lines 201–205: The identification of a unique 110-bp extension into rps19 in Crassulaceae is a notable finding. However, whether this extension is plesiomorphic or apomorphic is not addressed. Including context from the literature could provide valuable insight.

Lines 226–228: The revised phylogenetic tree shows minor differences in species placement relative to previous studies, yet the manuscript does not explore underlying biological factors such as morphology or biogeography. Addressing this could add depth to the analysis.

Lines 249–265: The discussion revisits codon usage and RNA editing but stops short of integrating these findings into a broader evolutionary framework for Sedum. This is an opportunity for further synthesis.

Lines 291–297: The discovery of a unique IR boundary pattern is a strength, but the potential applications or evolutionary implications of this feature are not discussed.

Lines 299–307: The section on polyphyly offers limited interpretation, simply reiterating the phylogenetic results. A deeper discussion of potential evolutionary drivers would enhance the study's impact.

Minor Comments:

Lines 25–27: In the Abstract, consider revising “our sequence data will enrich the organelle database and deepen our insights into the evolutionary history...” to a more concise version: “the generated sequence data enriches plastid resources and improves understanding of Sedum evolution.”

Line 69: It would be helpful to indicate whether voucher specimens or accession numbers were deposited in a herbarium or institutional repository.

Lines 85–87: You mention “six Crassulaceae plastomes from different genera, and eight Saxifragales plastomes...”—consider listing the taxa explicitly or providing a direct reference to Table S1.

Lines 127–129: The phrase “showing high similarities” should be corrected to “showing high similarity.”

Lines 174–176: When noting the RNA editing sites in ndhB and rpoC2, it would be helpful to clarify whether these numbers are consistent with patterns observed in other Crassulaceae taxa or represent novel findings.

Comments on the Quality of English Language

Overall, the manuscript is clearly written and suitable for publication, but improvements in fluency and cohesion would enhance its overall quality. A final revision by a native English-speaking expert in the field is strongly recommended before submission.

Reviewer 2 Report

Comments and Suggestions for Authors

The authors present eight new plastome sequences from Sedum and use them to explore phylogenetic relationships within Crassulaceae and Saxifragales. The topic is relevant, and the data are valuable, especially given the limited plastome resources available for this genus. The methods are generally sound, but a few steps need clarification, such as the parameters used in PREP-Cp and the phylogenetic tree construction strategy.

The results are informative and detailed, but more comparative context would be helpful—for example, how do these plastomes differ from previously published Sedum genomes? The rationale for selecting these eight species is not clearly explained. Some statements in the discussion feel a bit strong, such as the suggestion that the rps19 IR extension could serve as a species identification tool. It would also be useful to mention known limitations of plastome-based phylogenies, including hybridization and incomplete lineage sorting.

The English language needs polishing throughout to improve clarity and readability. Figures are clear, and the tree is well presented, though additional information on taxon sampling would strengthen the manuscript. Overall, this is a promising study that could be suitable for publication after substantial revisions to both content and language.

Comments on the Quality of English Language

A thorough review by a native speaker or professional editor is strongly recommended to improve clarity, flow, and scientific tone.

Round 2

Reviewer 2 Report

Comments and Suggestions for Authors

I recommand it for publication.